# Effect of Added Brewer’s Spent Grain on the Baking Value of Flour and the Quality of Wheat Bread

**DOI:** 10.3390/molecules27051624

**Published:** 2022-03-01

**Authors:** Anna Czubaszek, Agata Wojciechowicz-Budzisz, Radosław Spychaj, Joanna Kawa-Rygielska

**Affiliations:** Department of Fermentation and Cereals Technology, Wrocław University of Environmental and Life Sciences, 51-630 Wrocław, Poland; anna.czubaszek@upwr.edu.pl (A.C.); radoslaw.spychaj@upwr.edu.pl (R.S.); joanna.kawa-rygielska@upwr.edu.pl (J.K.-R.)

**Keywords:** wheat flour, dough and bread, brewer’s spent grain, baking value, bread quality and nutritional value

## Abstract

This study was undertaken to determine the effect of the partial replacement of wheat flour (WF) with barley brewer’s spent grain (BBSG) and barley-buckwheat brewer’s spent grain (BBSG + B) on dough quality and bread properties, including nutritional value. The contents of brewer’s spent grain (BSG) in the blend with wheat flour were 0, 10, and 20%. The quality of the flour blends was assessed with intermediate methods and based on laboratory baking. Analyses were also carried out to determine contents of basic nutrients and energy value. The replacement of part of wheat flour with BBSG and BBSG + B diminished gluten yield and deteriorated its quality (a decreased sedimentation value and stability, and increased dough softening). Changes were also observed in the starch-enzymatic system, resulting in a decreased falling number and maximum paste viscosity. Breads containing both BSG types featured higher yield and lower loaf volume. They had also higher contents of protein, dietary fibre, fat, and ash as well as a lower energy value compared to the wheat bread. Considering the organoleptic traits of breads, the 10% replacement of wheat flour with BSG is recommended in the blend. The BBSG + B was found to elicit more beneficial effects on bread properties than BBSG.

## 1. Introduction

Today, there is enormous political and social pressure around the world to reduce the amount of industrial waste. For this reason, many laboratories investigate the possibility of using various by-products to produce new, desired goods. Large amounts of by-products are generated by the food industry, many of which have a valuable composition and can still be used as raw material for further processing. The by-products generated by the brewing industry include waste yeast slurry, leavings, and brewer’s spent grain (BSG). Given the volume of global beer production (1.94 billion hectoliters in 2018) [1] and the fact that the production of 100 L of beer generates approximately 20 kg of BSG [2], it can be concluded that the amounts of BSG produced are considerable.

Authors [3] reported that BSG had a very valuable chemical composition and that its components could serve in the prophylaxis of lifestyle-related diseases. According to [2], BSG is rich in protein and fiber, which account for approximately 20% and 70% of its composition, respectively. A favorable feature of BSG protein is the high content of essential amino acids (about 30% of the total protein content), the major of which is lysine, considered to be the limiting amino acid in cereal products [4]. In turn, dietary fiber of BSG consists mainly of fibers—cellulose and hemicellulose, which are composed of xylose, arabinose, glucose, and lignin particles [5]. BSG is also rich in minerals, the most abundant of which include silicon, phosphorus, and calcium [4]. The nutritional value of BSG can vary greatly because its composition depends on many factors, including the genetic properties (genotype), type of barley (2 or 6-rows) and harvest date of the barley from which the malt was produced, the malting and mashing conditions, and the quality and type of additives used during the brewing process [6].

Due to the valuable composition of BSG, attempts have been made to use it to produce many food products, such as bread [4,5,7], cookies [8] or cereal snacks [9] Aside barley malt, other malted or non-malted raw materials are increasingly often used in brewing production as sources of sugar; these being wheat, rice, buckwheat or corn [10]. Among these raw materials, buckwheat (*Fagopyrum esculentum*) deserves special attention owing to its exceptional nutritional value. Its seeds are rich in protein featuring a high biological value approximating 90%. These properties are due to the high content of most essential amino acids, especially lysine, tryptophan, threonine, and the sulfur-containing amino acids [11]. Buckwheat products also provide compounds exhibiting antioxidant properties, including mainly phytosterols, tocols, flavonoids, and rutin. Authors [12] report that buckwheat contains 178 bioactive compounds.

The available literature lacks reports concerning the influence of barley-buckwheat brewer’s spent grain on the properties of wheat flour blends and bread made of them. With this in mind, a study was undertaken to compare and evaluate the effect of replacing a part of wheat flour with barley brewer’s spent grain and barley-buckwheat brewer’s spent grain on the baking value of flour blends determined with direct and indirect methods and on the nutrient composition of the bread made of these blends.

## 2. Results and Discussion

### 2.1. The Baking Value of Wheat Flour Blends with BSG

The quality of wheat dough and bread depends on the properties of the viscoelastic gluten network formed during dough making. Gluten proteins are responsible for the water absorption of the flour, dough development time and its resistance to mechanical processing, and they also affect CO_2_ retention in the dough [13]. Wheat flour intended for bread baking should have a wet gluten yield between 27 and 32% [14]. The results presented in Table 1 show that the wheat flour (WF) used in the present study was characterized by an appropriate wet gluten yield (29.0%). The amount of gluten eluted from the blends containing BBSG was lesser than from wheat flour and BBSG + B samples. The increase in the content of both BSGs from 10 to 20% caused no significant changes in this trait. Both, the quantity and quality of gluten are important from the standpoint of the technological process. According to [15], the quality of gluten can be determined using the Zeleny’s sedimentation index. Its average values determined by the above-mentioned authors were 35 cm^3^ for a flour with an average baking value and 41 cm^3^ for a strong flour. The value of Zeleny’s sedimentation index determined in the present study for wheat flour (37.5 cm^3^) indicates the good quality of its gluten proteins (Table 1). The addition of BBSG to the blends with wheat flour decreased the value of this index already at 10% BBSG content. In the samples containing BBSG + B, a significant difference was observed only with its 20% content.

The properties of wheat dough and bread also depend on the properties of the starch-amylase complex [16]. The falling number is an internationally recognized indicator of alpha-amylase activity in wheat flour [17]. According to [16], its values in wheat flour intended for bread making should reach exceed 220 s. The results presented in Table 1 indicate the optimal activity of wheat flour amylases. Both BSG types decreased the falling number of the flour blends, which was however still above 220 s. In the samples with BBSG, the falling number decreased with the increasing BSG content in the flour blend, whereas in those with BBSG + B, the mean values of this indicator did not differ significantly. Researchers [18] claim that a study of the rheological properties of a dough in a farinograph can provide practical information to enable the interpretation of dough quality and its behavior during processing. The results of farinographic analyses are presented in Figure 1. It was found that the WF flour was characterized by high water absorption (66.2%), and long dough development time and stability (4.1 and 6.5 min, respectively). This proves its good baking value. The softening of dough was significant and amounted to 120 FU, which indicates a low resistance of the dough made of it to mechanical treatment. As in the previous research with rye flour [19], the partial replacement of wheat flour with BBSG and BBSG + B caused a significant increase in the water absorption of the flour blend and an extension of the dough development time. The values of these traits increased with the increase in the content of both BSG in the flour blend. The BBSG 20% sample was characterized by the highest water absorption. In turn, the samples containing BBSG + B featured a longer dough development time than those with BBSG. Ref. [4] also showed that the addition of BSG to wheat flour affected the volume and rate of water binding in the dough. The changes in water absorption and development time of dough made of the flour with BSG, compared to the wheat dough, were probably due to an increased content of dietary fiber in the flour blend. Authors [18,20] claim that hemicellulose contained in BSG competes with gluten proteins for water and disrupts the process of a gluten network formation in the dough, which weakens its structure. The present study also showed dough weakening, because its stability decreased successively along with increasing contents of BBSG and BBSG + B (Figure 1b). Nevertheless, the softening of dough increased and was the same at both BSG inclusion levels (Figure 1c). The samples with BBSG were characterized by higher water absorption, shorter dough development time and stability, and higher dough softening compared to the samples containing BBSG + B. A similar correlation was observed upon rye flour replacement with BBSG and BBSG + B [19].

The amylographic examination enables evaluating the pasting process of wheat starch. Respective results are presented in Figure 2. There were no significant differences between the tested samples in terms of the initial and final gelatinization temperature. The pasting process started at 55.5–57.6 °C (BBSG 10% and BBSG 20%, respectively) and ended at 83.1–85.5 °C (WF and BBSG 20%, respectively). Our previous study [19] addressing the use of BBSG and BBSG + B as replacers for rye flour showed an increase in the initial gelatinization temperature, no changes in the final gelatinization temperature, and a significant reduction in paste viscosity under their influence. The present study also demonstrated a lower viscosity of BBSG and BBSG + B pastes compared to WF paste, and a decrease in the value of this parameter along with the increasing contents of both BSG types. The reason for the observed changes triggered by the increased content of both BSG types in the blends with wheat flour might be an increase in amylolytic activity, as indicated by a reduced falling number due to the BSG addition.

### 2.2. Quality of Wheat Bread Enriched with BSG

Table 2 shows the results of determinations of the physical properties of experimental breads. The results obtained allow concluding that the partial WF replacement with BBSG and BBSG + B increased the bread yield by 6–12% along with their increased content in the blend (WF-156%, 10% BBSG + B-162%, 20% BBSG-172%), which may be deemed beneficial in bakery due to economic concerns. As in our previous research [19] with rye bread, a higher yield was demonstrated for the breads containing BBSG than for those with BBSG + B. The observed correlations were probably due to the higher water-binding capacity of the flour blends, especially those with BBSG, than of WF, which was demonstrated in the farinographic analysis (Figure 1a).

The results presented in Table 2 indicate that replacing wheat flour with both BSGs resulted in a reduced bread volume per 100 g of flour and in the specific bread volume. There was also a tendency for bread volume decrease along with the increasing BSG content in the flour blend; however, the differences were not statistically significant. Other authors also reported a similar effect of BSG addition to wheat flour on the volume of bread made of it [4,7,21,22]. Ref. [23] demonstrated that the supplementation of wheat flour with non-glutinous flour resulted in lower bread volumes.

When assessing the bread crumb, it was found that bread color turned from light cream to brown under the influence of BBSG and BBSG + B addition (Appendix A). The darkest turned out to be the breads made of the flour with BBSG addition. The breads containing BBSG and BBSG + B had a more even porosity than wheat bread (Table 2, Appendix A). The 20% BBSG bread was rated the highest in this respect (7 points). Ktenioudaki et al. [21] made opposite observations regarding porosity changes. In the breadsticks studied by these authors, the crumb porosity deteriorated with an increasing BSG content. Presumably, the differences in the results obtained are due to the use of various experimental material and baking methods.

Table 2 presents the results of the organoleptic evaluation of breads. The enrichment of wheat bread with BBSG and BBSG + B resulted in decreased values of all assessed attributes. However, the statistical analysis of results demonstrated that the appearance, crust and crumb properties as well as taste and aroma of the breads with a 10% content of BBSG and BBSG + B did not differ significantly from those of the WF bread. The WF bread and breads with a 10% content of both BSGs were classified in quality level I, whereas those with a 20% content of BSGs were rated lower and classified in quality level III. The highest sum of points was recorded for the WF bread (30.9 points), followed by the BBSG + B bread (28.6 points). The lowest score was given to bread containing 20% BBSG (17.7 points). According to the panelists, it was less developed compared to the other breads, while its pores were very small and their walls were thick, which meant that its crumb was compact. Moreover, its taste and aroma were less acceptable than those of the WF loaves and those with a 10% content of both BSGs. Changes in the odor profile under the influence of the BSG additive were described by [9]. In turn, [7] found a significant decrease in the acceptability of bread even with a 5% content of BSG and a successive deterioration of this trait as the BSG content increased to 25%. Stojceska & Ainsworth [22] reported a decrease in the scores given for the texture of bread samples with an increase in BSG content from 10 to 30%.

### 2.3. Chemical Composition of Raw Material and Bread

Table 3 presents the contents of chemical components in wheat flour, BBSG, and BBSG + B. In comparison with the wheat flour, both BSG types had approx. 1.5 times higher protein content, approx. 12 times higher total dietary fiber content, approx. 3.5 times higher total fat content, and approx. 5 times higher ash content. On the other hand, the content of water and starch in BSG was about 2 times and 5–8 times lower, respectively, than in wheat flour. These results indicate that both BBSG and BBSG + B are rich sources of nutritionally valuable compounds. The high nutritional value of BSG was also evidenced by other authors [4,6,8,22,24,25].

The nutritional value of wheat bread enriched with BBSG and BBSG + B was assessed considering the content of basic nutrients and energy value. A significant increase in the moisture content of the bread crumb was found upon 20% wheat flour replacement with both BSG types (Table 3). The contents of total protein, dietary fiber, and ash increased significantly with the increase in BBSG and BBSG + B contents in the bread. The breads containing BBSG + B had higher protein and lipid contents than these with BBSG, which was expected considering their contents in the raw material. Compared to the WF bread, the ash content was significantly higher only in 20% BBSG, and 10% and 20% BBSG + B breads. In turn, starch content decreased with the increase in the BBSG and BBSG + B inclusion levels, with both BSGs causing similar changes in its value. A significant reduction in energy value was also demonstrated with a 20% content of BBSG and 10 and 20% contents of BBSG + B in the flour blends. The results of the present study confirm the findings from research of other authors [7,9,10,21], who demonstrated increased contents of protein and dietary fiber and a decreased energy value of the breads after BSG inclusion into their recipe.

### 2.4. Principal Component Analysis (PCA)

The PCA allowed determining correlations between the features characterizing the protein and amylase-starch complexes of flour and the quality attributes of the breads (Figure 3). The significant values of Pearson’s correlation coefficients determined in the study are included in the Appendix A.

The sum of the principal components determined for the protein complex and quality attributes of the bread was 95.17% (PC1: 72.87%, PC2: 22.30%) (Figure 3a). It was found that the yield of bread (YB) was negatively correlated with the Zeleny’s sedimentation index (ZSI) and positively with dough development time (DDT) (Appendix A). The mentioned features characterizing the properties of the protein complex also affected both parameters of the bread volume (BV, SV), but the correlations between them were opposite to those noted for the bread yield. Our previous research into the influence of BBSG and BBSG + B on the properties of blends with rye flour [19], also demonstrated a positive correlation between dough development time and bread yield and a negative correlation with specific volume. The present study showed a correlation between crumb porosity (POC) and Zeleny’s sedimentation index (ZSI, r = −0.893), water absorption (WA, r = 0.979) and stability of dough (STAB, r = −0.939). Wet gluten did not correlate with any of the bread characteristics tested.

When assessing the correlations between the features describing the properties of the amylase-starch complex and the quality of bread based on the PCA analysis, it was found that the value of the first principal component was 68.62% (PC1) and that of the second principal component was 18.59% (PC2) (Figure 3b). There was a significant positive correlation between the volume of bread (BV) and the falling number (FN) (Appendix A). It was also shown that bread porosity positively correlated with the final gelatinization temperature (FT), and negatively—with the maximum paste viscosity (MV). There was no significant correlation between the yield of bread (YB) and the features describing the amylose-starch complex, whereas the initial gelatinization temperature (IT) did not correlate with any of the quality attributes of the bread.

Based on the calculated values of Pearson’s correlation coefficients, a correlation was also demonstrated between the quality attributes of the bread. Yield of bread (YB) negatively correlated with bread volume (BV, SV), while the overall score in the organoleptic assessment negatively correlated with bread yield and positively with its volume.

## 3. Material and Methods

### 3.1. Material

Commercial wheat flour type 650 (WF) was from mill company Dolnośląskie Młyny S.A. (Ujazd Górny, Poland). Two types of brewer’s spent grains (BSG): barley (BBSG) and barley-buckwheat (BBSG + B) came from the brewery Browar Stu Mostów (Wrocław, Poland). BBSG and BBSG + B were preserved by drying at a temperature of 55 °C for 3 h in the UF110 Plus dryer (Memmert GmbH + Co, Schwabach, Germany). Before starting the research, both BSG were ground in a laboratory hammer mill WŻ1 (ZBPP, Bydgoszcz, Poland). The granulation of milled BSG was below 250 μm. Wheat flour and ground BBSG and BBSG + B were used to prepare blends with BSG contents of 10 and 20%. Wheat flour served as the control. The other ingredients used in the production of bread were evaporated iodized salt produced by Cenos Sp. z o.o. (Września, Poland) and pressed baker’s yeast produced by Lesaffre Polska S.A. (Wołczyn, Poland), bought in the local market.

### 3.2. Technological Analyses of Flour and Its Blends with BSG

Wheat flour and its blends with BBSG and BBSG + B were determined for: wet gluten content [26] and falling number [27]. The quality of wet gluten was determined based on the Zeleny’s sedimentation index [28]. Properties of pastes were evaluated using an amylograph (Brabender OHG, Duisburg, Germany) according to AACC Methods 22–10 [29]. The initial and final gelatinization temperature and the paste maximum viscosity were determined from the amylograms. The rheological properties of dough were analyzed acc. to the AACC Methods 54–21 using a farinograph (Brabender OHG, Duisburg, Germany) [29]. The farinographic analysis allowed determining: water absorption of flour (% compared to flour used), dough development time, stability and softening of dough (FU—farinograph units).

### 3.3. Laboratory Baking

Bread was baked under laboratory conditions with the single-stage method, according to the following recipe: wheat flour or its blend with BSG—250 g, salt—3.8 g, and compressed yeast—7.5 g. The dough was prepared in a farinograph mixer (bowl for 300 g of flour) by adding tap water having a temperature of 30 °C, in a volume ensuring dough consistency of 300 FU (182–225 cm^3^), at mixing time 5 min. The doughs were put in the molds (8.5 × 8.5 cm^2^ at the base, 13 × 13 cm^2^ at the top edge, 10.5 cm in height), greased with oil. The molds were placed in a fermentation chamber (Eka, Padova, Italy) for 90 min. During fermentation, the dough was degassed after 60 and 90 min and left for the final fermentation (45–50 min). The breads were baked in two replications in a GT800 electric furnace (IBIS, Szubin, Poland), at a temperature of 245 °C for 30 min, with steaming in the first 3 min of baking.

### 3.4. Physical Properties of Bread

After baking, the breads were taken out of the molds and cooled for about 2 h at 20 °C. After cooling, they were weighed and their volume was measured with an SA-WY bread volumeter (ZBPP, Bydgoszcz, Poland) filled with millet grain. Analyses were performed at least in duplicate. Bread weight was used to determine bread yield in respect of the weight of flour used for baking. Bread volume was converted as bread volume per 100 g of flour and specific volume (bread loaf volume divided by bread loaf weight). The crumb porosity was assessed according to the Dallmann scale [30].

### 3.5. Organoleptic Evaluation of Bread

A team of 10 trained panelists conducted the organoleptic assessment of bread according to the Polish Standard [31]. All study participants consciously agreed to perform the analysis. The assessment was made for such attributes as: external appearance of a bread loaf, crust characteristics (color, external appearance, elasticity, crispness, and thickness), crumb characteristics (elasticity, porosity, moisture, and viscosity perceptible to the touch), taste, and aroma. During bread assessment, its quality attributes were compared with the description provided in the Polish Standard and scored a certain number of points (from 2 to 6 per attribute). However, in the assessment of each quality attribute, 0 points were scored for deviations that did not disqualify the bread, and −35 points for deviations that disqualified the bread. Bread quality was evaluated based on the average total number of points given by the panelists to each bread type. The quality level was determined based on the following classification: 32–28 points—level I, 27–23 points—level II, 22–18 points—level III, 17–0 points—level IV, below 0 points—disqualification. According to the adopted classification, the bread rated as level I was very well developed; and was characterized by a smooth crust of appropriate thickness and color. Its crumb with an even and fine porosity was very elastic, and its flavor and aroma were mild, typical of wheat bread. Level IV bread was accepted by consumers, sufficiently well-risen with slight deviations in shape. Its crust was slightly uneven in color, and the crumb was sufficiently elastic with a slightly uneven porosity and color. The taste and aroma might slightly differ from that of wheat bread.

### 3.6. Chemical Composition of Flour, Flour Blends and Bread

Breads were determined for: total protein content—with the Kjeldahl method [32] using a Foss Tecator Kjeltec 2400 analyzer (Foss, Hilleroed, Denmark) (N × 5.7 for wheat flour and all breads, N × 6.25 for BSG), ash content—AACC Method 46.11A, [29], and total dietary fiber content—AOAC Method 985.29, [33] using total dietary fiber assay kits TDF-100A-1KT and TDF-C10 (Sigma-Aldrich, Saint Louis, MO, USA). Moreover, the content of starch was determined polarimetrically with the Ewers method [34], and the lipid content—with the Soxhlet method—AOAC, Method 935.38 [33]. The energy value was calculated using the conversion factors in accordance with [35], Annex XIV. The samples were analyzed at least in duplicate, and the results are expressed on a dry matter (d.m.) basis.

### 3.7. Statistical Analysis

The results presented are mean values ± standard deviation (SD). Statistical analyses of results obtained for WF, BBSG, BBSG + B, blends thereof, and breads with 10 and 20% content of BBSG and BBSG + B were conducted with the one-way analysis of variance (ANOVA). Significant differences (*p* ≤ 0.05) between the mean values were determined using the Duncan’s Multiple Range Test. The Principal Component Analysis (PCA) was performed to determine correlations between the quality attributes of bread and parameters indicating the quality of protein and amylose-starch complexes. Statistical calculations were carried out at *p* = 0.05 using the Statistica 13.0 data analysis software system (TIBCO Software Inc., Kraków, Poland, 2017).).

## 4. Conclusions

The partial replacement of wheat flour with BSG resulted in a decreased content of wet gluten and its deteriorated quality assessed based on the Zeleny’s sedimentation index and farinographic analysis. The falling number and viscosity of the flour pastes decreased along with the increase in the BSG content in the flour blend. The BBSG + B contributed to unfavorable changes in the baking value of the flour blends to a lesser extent than BBSG. An increase in bread yield and a decrease in the volume of loaves were found in laboratory baking, under the influence of both BSG types. Considering the nutritional value, the breads containing both BSGs turned out to be superior over wheat bread. With the increase in the content of both BSG types in flour blends, the contents of protein, dietary fiber, fats and ash significantly increased and the energy value decreased, while BBSG + B turned out to be a better additive than BBSG. In the organoleptic assessment, the breads with 10% BBSG and BBSG + B were classified as of quality level I, similarly to the control bread. The breads with BBSG + B were scored higher than these with BBSG.

## Figures and Tables

**Figure 1 molecules-27-01624-f001:**
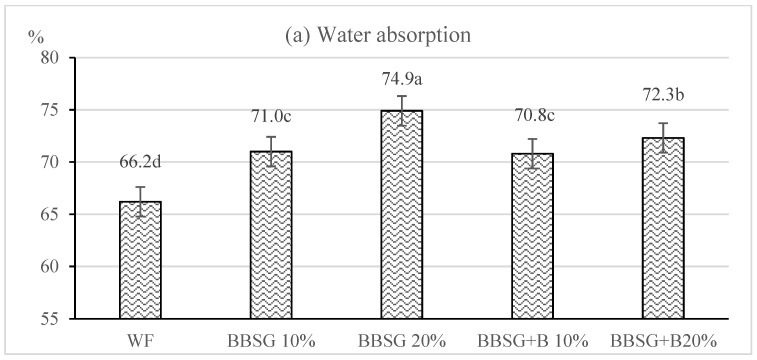
Farinographic traits of wheat flour and its mixtures with BBSG and BBSG + B: (**a**) water absorption; (**b**) dough development time and stability; (**c**) softening of dough. a, b, c, d—mean values denoted with different letters are significantly different at *p* ≤ 0.05.

**Figure 2 molecules-27-01624-f002:**
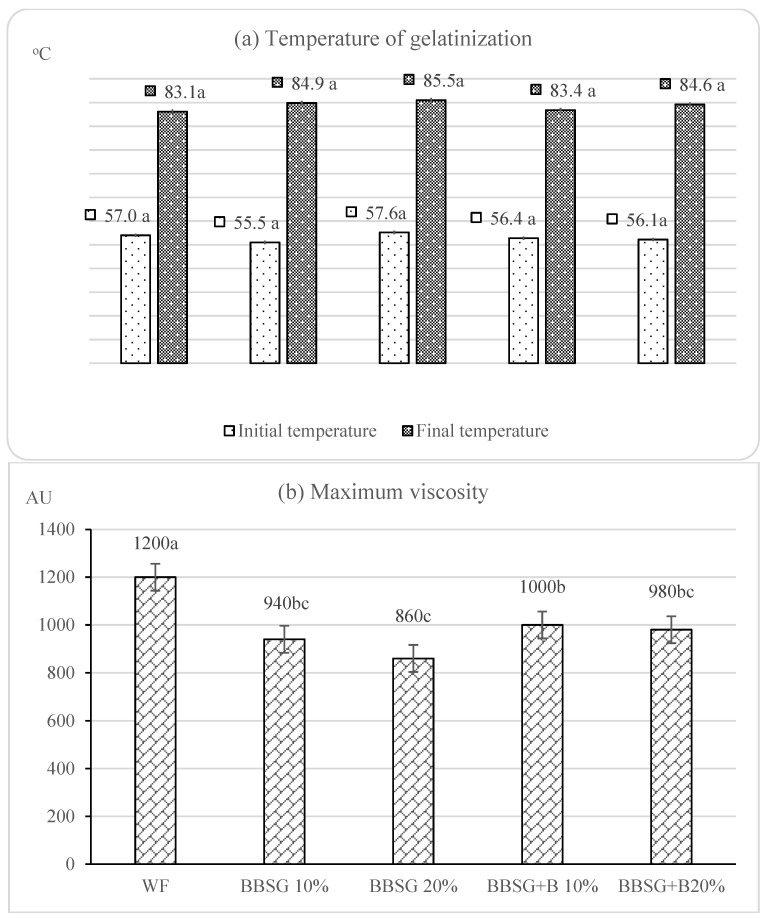
Amylographic traits of wheat flour and its mixtures with BBSG and BBSG + B: (**a**) temperature of gelatinization, (**b**) maximum viscosity. a, b, c—mean values denoted with different letters are significantly different at *p* ≤ 0.05.

**Figure 3 molecules-27-01624-f003:**
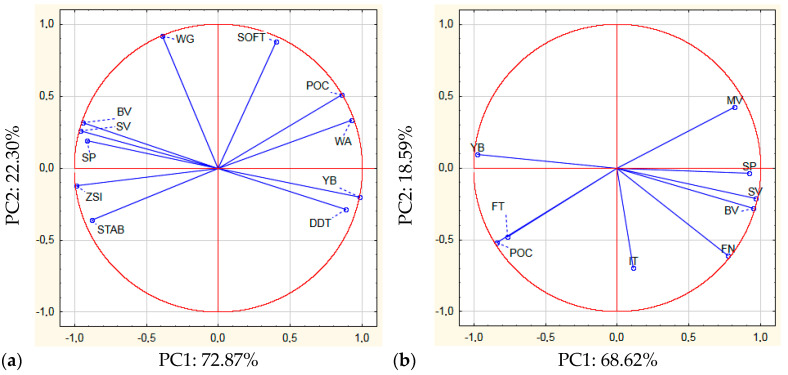
Principal component analysis between features of the protein complex (**a**), amylose-starch complex (**b**) and quality traits of bread. YB -Yield of bread, BV-bread volume per 100 g flour, SV-specific volume, POC-porosity of crumb Dallmann scale, SP-sum of points, WG-wet gluten, ZSI-Zeleny sedimentation index, WA-water absorption of flour, DDT-dough development time, STAB-stability of dough, SOFT-degree of dough softening, FN-falling number, IT-initial gelanization temperature, FT-final gelanization temperature, MV-maximum viscosity.

**Table 1 molecules-27-01624-t001:** Quality tratis of wheat flour and it blends with BBSG and BBSG + B.

Sample	Wet Gluten [%]	Zeleny’s Sedimentation Index [cm^3^]	Falling Number [s]
WF	29.0 ^a^ ± 0.2	37.5 ^a^ ± 0.7	336 ^a^ ± 1
BBSG 10%	24.5 ^b^ ± 0.8	33.0 ^b^ ± 0.0	323 ^b^ ± 7
BBSG 20%	22.2 ^b^ ± 0.8	29.5 ^c^ ± 0.7	286 ^c^ ± 6
BBSG + B 10%	28.3 ^a^ ± 0.5	32.0 ^b^ ± 0.0	318 ^b^ ± 2
BBSG + B 20%	27.0 ^a^ ± 0.5	30.0 ^c^ ± 0.0	326 ^ab^ ± 1

a, b, c—mean values denoted in columns with different letters are significantly different at *p* ≤ 0.05.

**Table 2 molecules-27-01624-t002:** Quality traits of bread.

Traits			WF	BBSG		BBSG + B	
				10%	20%	10%	20%
Physical traits							
Yield of bread		[%]	156 ^c^ ± 2	164 ^b^ ± 4	172 ^a^ ± 8	162 ^bc^ ± 3	168 ^ab^ ± 3
Volume of bread per 100 g flour		[cm^3^]	574 ^a^ ± 23	500 ^ab^ ± 27	434 ^b^ ± 62	518 ^ab^ ± 59	492 ^ab^ ± 72
Specific volume		[cm^3^/g]	3.68 ^a^ ± 0.17	3.05 ^bc^ ± 0.23	2.54 ^c^ ± 0.48	3.20 ^ab^ ± 0.42	2.94 ^bc^ ± 0.47
Porosity of the crumb in Dallmann scale		[points]	2 ^b^ ± 0	5 ^ab^ ± 0	7 ^a^ ± 0	4 ^ab^ ± 0	5 ^ab^ ± 0
Organoleptic assessment							
Appearance		max 5 points	5.0 ^a^ ± 0.0	4.6 ^a^ ± 0.6	1.8 ^c^ ± 1.5	4.8 ^a^ ± 0.5	3.4 ^b^ ± 0.6
Crust	Colour	max 3 points	3.0 ^a^ ± 0.0	2.7 ^ab^ ± 0.5	2.4 ^ab^ ± 0.7	2.7 ^ab^ ± 0.5	2.0 ^b^ ± 1.1
	Thickness	max 4 points	4.0 ^a^ ± 0.0	3.8 ^a^ ± 0.4	3.1 ^c^ ± 0.9	3.8 ^ab^ ± 0.5	3.1 ^bc^ ± 0.8
	other features	max 4 points	4.0 ^a^ ± 0.0	3.8 ^a^ ± 0.4	2.8 ^b^ ± 0.5	3.8 ^a^ ± 0.5	2.9 ^b^ ± 0.4
Crumb	Elasticity	max 4 points	4.0 ^a^ ± 0.0	3.8 ^a^ ± 0.5	2.4 ^c^ ± 0.5	3.8 ^a^ ± 0.5	3.0 ^b^ ± 0.8
	Porosity	max 3 points	2.4 ^ab^ ± 0.5	2.6 ^a^ ± 0.5	1.6 ^b^ ± 1.1	2.4 ^ab^ ± 1.1	2.4 ^ab^ ± 0.5
	other features	max 3 points	2.6 ^a^ ± 0.5	2.9 ^a^ ± 0.4	1.9 ^b^ ± 0.6	2.9 ^a^ ± 0.4	1.9 ^b^ ±0.9
Taste and smell		max 6 points	5.9 ^a^ ± 0.4	3.8 ^bc^ ± 2.4	1.8 ^c^ ± 1.6	4.6 ^ab^ ± 2.0	2.4 ^c^ ± 2.2
Sum of points			30.9 ^a^ ± 1.1	27.8 ^a^ ± 3.0	17.7 ^c^ ± 3.1	28.6 ^a^ ± 2.9	21.3 ^b^ ± 3.7
Quality level			I	I	III	I	III

a, b, c—mean values denoted in columns with different letters are significantly different at *p* ≤ 0.05.

**Table 3 molecules-27-01624-t003:** The content of nutrients in raw material and bread.

Sample	Moisture	Total Protein	Starch	Total Dietary Fiber	Lipids	Ash	Energy Value
	[g/100 g d.m.]	[g/100 g d.m.]	[g/100 g d.m.]	[g/100 g d.m.]	[g/100 g d.m.]	[g/100 g d.m.]	[kcal]
Raw material							
WF	12.6 ^a^ ± 0.25	13.5 ^c^ ± 0.1	72.5 ^a^ ± 0.1	4.3 ^b^ ± 0.5	2.2 ^c^ ± 0.02	0.64 ^c^ ± 0.05	-
BBSG	6.0 ^b^ ± 0.27	21.0 ^b^ ± 0.0	13.4 ^b^ ± 0.2	54.9 ^a^ ± 0.3	7.3 ^b^ ± 0.03	2.74 ^b^ ± 0.01	-
BBSG + B	6.5 ^b^ ± 0.27	25.0 ^a^ ± 0.3	8.6 ^c^ ± 0.1	52.5 ^a^ ± 0.3	8.9 ^a^ ± 0.01	3.44 ^a^ ± 0.03	-
Bread							
WF	43.1 ^b^ ± 1.1	13.4 ^e^ ± 0.0	75.5 ^a^ ± 0.1	5.0 ^c^ ± 0.2	2.56 ^e^ ± 0.0	1.25 ^c^ ± 0.1	225 ^a^ ± 4.2
BBSG 10%	46.1 ^ab^ ± 2.7	14.6 ^d^ ± 0.1	70.4 ^b^ ± 0.5	10.0 ^b^ ± 0.1	2.66 ^d^ ± 0.1	1.32 ^bc^ ± 0.0	216 ^ab^ ± 1.5
BBSG 20%	48.9 ^a^ ± 0.1	15.7 ^b^ ± 0.0	63.9 ^c^ ± 0.2	14.9 ^a^ ± 0.1	3.36 ^b^ ± 0.0	1.51 ^ab^ ± 0.0	206 ^b^ ± 2.9
BBSG + B 10%	42.8 ^b^ ± 0.4	15.0 ^c^ ± 0.1	69.4 ^b^ ± 0.8	10.6 ^b^ ± 0.0	2.86 ^c^ ± 0.0	1.50 ^ab^ ± 0.0	204 ^b^ ± 8.5
BBSG + B 20%	44.4 ^b^ ± 0.3	16.4 ^a^ ± 0.0	63.0 ^c^ ± 0.1	14.8 ^a^ ± 0.9	3.92 ^a^ ± 0.0	1.68 ^a^ ± 0.1	185 ^c^ ± 0.5

a, b, c, d—mean values denoted in columns with different letters are significantly different at *p* ≤ 0.05.

## Data Availability

Data will be made available upon request directed to the corresponding author. Proposals will be reviewed and approved by the investigators and collaborators based on scientific merit.

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
