# Peer review of "Effect of Added Brewer’s Spent Grain on the Baking Value of Flour and the Quality of Wheat Bread"

_molecules, 2022, doi:10.3390/molecules27051624_

Round 1

Reviewer 1 Report

The following criteria stood as the basis for my decision:

- the use of waste and by-products of the beer industry in other branches of the food industry has preocupated numerous researchers throughout the years (K.M. Lynch, et.al. Brewer's spent grain: a review with an emphasis on food and health, J.Inst. Brew., 2016, S.I. Mussatto, et.al. Brewer's spent grain: generation, characteristics and potential applications, Journal of Cereal Science, 2006,  etc.). So much so that it is difficult to identify novel ideas in this field. I could not consider as innovative the simple modification of a cereal type/percentage in the composition of the by-product.

- it falls within the trends to repurpose by-products, as a part of circular economy principles, however previous studies have shown that the taste and texture of the baked goods could be severely perturbated and prevents the addition of such cereal by-products in higher proportions

- the investigation methods employed in the paper are also well known and not very recent (e.g. farinograph) and complete rheological analyses are missing

- on the other hand, the paper is in essence well written, coherently follows the proposed objectives and aim the data is analysed statistically and well discussed

To sum up, there were insufficient grounds for the rejection of this paper, while any improvements would imply a different analytical approach (such as enlarging the sensory analysis group, determining the amino acid content of the sample breads, performing further rheological analyses etc.) and could not be addressed with a revision of the paper.

Reviewer 2 Report

The present work evaluates the effect of partial replacement of wheat flour with barley brewer´s spent grain (BBSG) and barley-buckwheat brewer´s spent grain (BBSG+B). The flour replacement levels used in this work were: 0, 10, and 20%. Several variables were measured regarding chemical composition, organoleptic traits, and rheological properties.

The manuscript is well written, well structured, and denotes a high-quality work in general terms. In this type of research, I would like to highlight the number of variables evaluated, which involve hard work, time, and economic costs. The design is appropriate, and the research is perfectly reproducible. One of the main strengths of the current study is the results section, which is excellently supported by the discussion. The bibliography cited is in accordance. Furthermore, the authors have experience and performed several previous works on the subject (e.g., baking properties of rye bread).

An interesting point of the work is that it showed novel information regarding baking quality using alternative grains such as buckwheat (Fagopyrum esculentum). Added to this, the relevance of this crop worldwide has been increasing in the last years because it is a gluten-free grain. Therefore, the present work is novel and makes significant contributions regarding baking quality and the reuse of BSG.

In order to improve the quality of the manuscript, some minor points were marked in the attached file and below:

-Page 2, line 47: please, replace “variety” with “genotype.” Variety is ambiguous and refers to the botanical variety (taxonomy).

-Page 2, line 49: please, add the type of barley (2 or 6-rows) because it is a very important factor in determining the nutritional value (e.g., protein content).

-Page 2, line 54: please, add the scientific name of the buckwheat.

-Page 2, lines 70-73: Very long sentence. Please, break in two.

-Page 8, line 253: Wet gluten is a percentage, as well described by the authors before. Please, remove the word “yield” after “wet gluten.”

-Page 9, line 276: Please, unify along the complete manuscript the abbreviation of “Bread yield.” Sometimes appears as “BY” and other times appears as “YB” (see PCA graph and page 9, line 260).

-Page 9, line 285: Correct the expression “°C “ (superscript letter).

-Page 10, line 296: What do the authors mean by "the quality of wet gluten"? I suggest starting the sentence directly with "Wet gluten was determined […]”.

Some suggestions for future works (not necessary to include them now, just suggestions):

(i) Statistical analysis is correct. However, I  would have preferred the SEM to be shown (instead of the SD) since it includes the number of samples evaluated. Furthermore, I would have preferred the Tukey test because the Duncan test is more flexible and may show some significance that otherwise would not.

(ii) Regarding figures (Fig. 1, 2, and 3), although they are appropriate, do not show good quality. I suggest using ggplot2 (R language) or GraphPad Prism to get higher quality figures. As to supplemental files and tables, they are correct.

Therefore, I consider that the current version of the manuscript would be accepted for publication in Molecules, after some minor modifications and according to the editorial board's decision.

Reviewer 3 Report

This study was conducted to investigate the potential application of BBSG and BBSG+B in the development of bread. The manuscript is generally well written and structured. However, there are a few aspects that required attention:

  1. Abstract - it is recommended to include significant results obtained from the study.
  2. Results and discussion - CO2 (line 73) should use subscript. Line 122 to 130 should be discussed earlier before line 96. Further discussion needs to be included that contributed to the decrease in Falling Number. Statement in lines 210 to 211 needs to be rechecked. Line 266 - should be amylose instead of amylase. 
  3. Materials and methods - The amount of protein content of WF (line 282) needs to be included. The degree symbol (line 285) needs to use superscripts.  The details for Falling Number protocols is required (line 296). The brand of fermentation chamber (line 313) needs to be included. 
